# Recent Advancements in Understanding the Role and Mechanisms of Angiopoietin-like Proteins in Diabetic Retinopathy

**DOI:** 10.3390/metabo15060352

**Published:** 2025-05-26

**Authors:** Xinling Zhang, Dongang Liu, Yuting Qiu, Ruiyao Hu, Shiyu Chen, Yue Xu, Chenyan Qian, Lailing Du, Jinghua Yuan, Xiaoping Li

**Affiliations:** Key Laboratory of Artificial Organs and Computational Medicine in Zhejiang Province, Shulan International Medical College, Zhejiang Shuren University, Hangzhou 310015, China; xinlingz_201803@zjsru.edu.cn (X.Z.); 202211002127@stu.zjsru.edu.cn (D.L.); 202211002114@stu.zjsru.edu.cn (Y.Q.); 202211001205@stu.zjsru.edu.cn (R.H.); 202211002102@stu.zjsru.edu.cn (S.C.); 202211003324@stu.zjsru.edu.cn (Y.X.); 202310301412@stu.zjsru.edu.cn (C.Q.); dulailing@zjsru.edu.cn (L.D.); 601660@zjsru.edu.cn (J.Y.)

**Keywords:** angiopoietin-like protein, diabetic retinopathy, molecular mechanism, early diagnosis, targeted therapy

## Abstract

Angiopoietin-like proteins (ANGPTLs) represent a family of secreted glycoproteins that are extensively expressed in vivo and are integral to various pathophysiological processes, including glucose and lipid metabolism, stem cell proliferation, local inflammation, vascular permeability, and angiogenesis. Particularly interesting is ANGPTL4, which has been identified as a significant factor in the development and progression of diabetic retinopathy (DR), thus becoming a central focus of DR research. ANGPTLs modulate metabolic pathways, enhance vascular permeability, and facilitate pathological angiogenesis, in addition to causing intraocular inflammation. As promising molecular targets, ANGPTLs not only serve as biomarkers for predicting the onset and progression of DR but also present therapeutic potential through antibody-based interventions. This paper discusses the pathogenesis of DR and the potential applications of ANGPTLs in early diagnosis and targeted therapy. It provides references for advancing precision diagnosis and personalized treatment strategies through more profound ANGPTLs research in the future.

## 1. Introduction

Diabetic retinopathy (DR) is the most prevalent microvascular issue affecting the eye health of people with diabetes. It is a leading cause of vision loss and blindness among older adults with diabetes worldwide [1,2]. In 2020, around 103.12 million adults globally were impacted by DR, and forecasts predict this number will rise to 160 million by 2045 [1,3]. The main pathological process in DR is linked to changes in the microvasculature, with inflammation and retinal neurodegeneration also identified as significant early pathological changes related to retinal damage in DR [4]. Emerging research underscores the multifactorial nature of DR, with various key molecules and signaling pathways implicated in its pathogenesis. For example, vascular endothelial growth factor (VEGF) and angiopoietin (Ang) have been demonstrated to facilitate endothelial cell proliferation, angiogenesis, and alterations in vascular permeability in DR. Therapeutic interventions targeting VEGF and Ang have achieved notable clinical success. Recent research has increasingly underscored the significance of angiopoietin-like proteins (ANGPTLs), with a particular emphasis on ANGPTL4, in a variety of pathophysiological processes, including glucose and lipid metabolism, inflammation, angiogenesis, tumor metastasis, apoptosis, and stem cell proliferation [5].

ANGPTLs are recently discovered secreted glycoproteins that, although they have some structural similarities and domains in common with angiopoietins, have unique roles partly because they cannot bind to tyrosine kinase receptors 1 and 2. ANGPTL4 has emerged as a prominent area of interest in DR research. Research indicates that the levels of ANGPTL4 in the vitreous body and serum are markedly higher in PDR patients than in those with idiopathic macular hole (IMH).

ANGPTL4 and VEGF levels show a significant correlation in the vitreous body and serum of PDR patients, suggesting that ANGPTL4 may serve as a new therapeutic target [6,7]. Research indicates that the expression levels of serum ANGPTL3 positively correlate with the stage of diabetic retinopathy (DR) in individuals with type 2 diabetes mellitus (T2DM), implying its potential use as a biomarker for tracking the progression of DR [6]. ANGPTL8 expression is markedly increased in the vitreous body of PDR patients, and its serum levels are significantly associated with DR [8,9]. Furthermore, ANGPTLs are ubiquitously expressed throughout the body and are involved in regulating glucose and lipid metabolism, modulating inflammatory responses, promoting angiogenesis, enhancing vascular permeability, inhibiting apoptosis, and regulating stem cell activity, all of which are crucial for maintaining normal physiological functions. This paper aims to bring together the existing knowledge on the roles and mechanisms of ANGPTLs in DR, providing new perspectives on the disease’s pathogenesis, clinical diagnosis, and treatment. In addition, ANGPTL3, ANGPTL4, and ANGPTL8 may be biomarkers for DR.

## 2. ANGPTL Family Protein Member

The ANGPTL family consists of eight glycoprotein members, with ANGPTL1-7 distinguished by the presence of an N-terminal coiled-coil domain (CCD) and a C-terminal fibrinogen-like domain (FLD). In contrast, ANGPTL8 is an atypical member, lacking both the CCD and FLD. ANGPTLs are expressed throughout the body and play a vital role in sustaining normal bodily functions. They are also involved in several disease-related processes [5]. Research indicates a strong association between ANGPTLs and DR. Notably, ANGPTL4 has been shown to upregulate VEGF expression in the eye, thereby promoting ocular angiogenesis and inflammation in DR patients. Elevated levels of ANGPTL2, ANGPTL4, and ANGPTL8 have been detected in the vitreous humor of patients with proliferative diabetic retinopathy (PDR), while serum ANGPTL3 levels have been found to correlate positively with DR severity. These findings suggest that ANGPTL proteins may play a significant role in the onset and progression of DR. As shown in Table 1.

## 3. The Occurrence and Progression of DR Involve ANGPTL

Currently, research on the relationship between ANGPTL4 and DR has become more comprehensive and relatively well-defined. In contrast, ANGPTL2, ANGPTL3, ANGPTL4, and ANGPTL8 have different degrees of correlation with the onset and progression of DR, though the precise mechanisms remain elusive.

### 3.1. ANGPTL2 and DR

ANGPTL2 was initially identified in the vitreous of DR patients in 2020. Its expression is not only elevated in the vitreous of patients with PDR, but it may also be associated with the progression of PDR [43]. Keles et al. observed that ANGPTL2 expression levels were increased in the vitreous of patients with active PDR [43]. Moreover, in PDR patients with different complications, a notable rise in ANGPTL2 expression was observed in eyes with fibrovascular traction detachment, indicating a possible connection between ANGPTL2 expression and the advancement of PDR.

### 3.2. ANGPTL3 and DR

Additionally, Yu et al. [11] conducted a study involving 1192 patients with type 2 diabetes mellitus (T2DM), where serum levels of ANGPTL3, ANGPTL4, C-reactive protein, vascular adhesion molecule-1, and intracellular adhesion molecule-1 were quantified using enzyme-linked immunosorbent assay. The findings indicated that, compared to a normal control group, T2DM patients exhibited elevated serum levels of ANGPTL3, which significantly increased the risk of DR. Furthermore, in comparison to non-proliferative diabetic retinopathy (NPDR) patients, elevated serum ANGPTL3 levels in T2DM patients were associated with a heightened risk of PDR, suggesting that ANGPTL3 may play a crucial role in the progression of DR.

### 3.3. ANGPTL4 and DR

ANGPTL4 (angiopoietin-like 4) is a multifunctional glycoprotein that regulates lipid metabolism, angiogenesis, and inflammation. Previous studies have demonstrated that the expression level of ANGPTL4 is significantly elevated in the vitreous body and serum of patients with PDR, and this elevation is positively correlated with the expression level of VEGF [44]. Furthermore, the levels of ANGPTL4 in the vitreous and serum of PDR patients show a positive correlation with serum triglycerides (TG) and a negative correlation with high-density lipoprotein cholesterol (HDL-C). ANGPTL4 modulates lipoprotein lipase activity, affecting lipid accumulation in retinal cells [45]. These findings suggest that ANGPTL4 is significantly associated with lipid metabolism disorders and intraocular pathological changes in PDR patients, indicating that ANGPTL4 may play a crucial role in the onset and progression of PDR. Additionally, several studies have reported increased levels of ANGPTL4 in the aqueous humor of DR patients, with a positive correlation between ANGPTL4 levels and the severity of DR [46,47,48]. ANGPTL4 inhibits VEGF-induced retinal vascular leakage by disrupting integrin signaling [49]. Teng Yue et al. further validated that the expression level of ANGPTL4 in the aqueous humor of patients with DR was elevated, as determined by liquid flow chip technology. This elevation was more pronounced in patients with PDR compared to those with NPDR. Concurrently, the study identified a positive correlation between ANGPTL4 and interleukin (IL)-6, IL-8, monocyte chemoattractant protein-1, the vascular endothelial growth factor (VEGF) family, and the platelet-derived growth factor family. ANGPTL4 was a decreased ligand–receptor expression and improved retinal function as a potential therapeutic target against DR [50]. The instability of serum ANGPTL4 expression, influenced by various factors, may explain this discrepancy. Thus, the essential role of ANGPTL4 in DR’s development and progression requires additional research.

### 3.4. ANGPTL8 and DR

In a cross-sectional study, Wang et al. [51] examined serum ANGPTL8 expression levels among various groups, including patients with type 2 diabetes mellitus (T2DM) with and without DR, healthy controls, newly diagnosed T2DM patients prior to treatment, and T2DM patients undergoing hypoglycemic therapy. Their findings indicated that serum ANGPTL8 expression levels were elevated in T2DM patients receiving hypoglycemic treatment. Furthermore, an increase in serum ANGPTL8 expression was positively correlated with the incidence of DR in T2DM patients. Conversely, another cross-sectional study by Fang et al. demonstrated that ANGPTL8 was a significant and independent variable associated with retinopathy by comparing serum ANGPTL8 levels in T2DM and DR patients [8]. Recent research has confirmed these results, indicating a notable increase in serum ANGPTL8 levels among DR patients [52]. Additionally, the expression levels of ANGPTL8 and VEGF in the vitreous and serum of patients with PDR were higher than those in patients with idiopathic macular hole (IMH), with a positive correlation between these two factors [9].

## 4. Molecular Mechanisms of ANGPTL in DR

The pathogenesis of DR is intricately linked to disorders in glucose and lipid metabolism, alterations in vascular permeability, angiogenesis, and intraocular inflammation. Current research posits that hyperglycemia and hyperlipidemia are risk factors for DR, directly facilitating its onset and progression. The primary pathological changes in DR include increased vascular permeability, pathological angiogenesis, and fibrosis. Although various ANGPTLs are implicated in the pathological changes associated with DR, no studies have yet confirmed the involvement of ANGPTL in ocular fibrosis. In addition, intraocular inflammation plays a key role in DR and runs through the whole process of DR (Figure 1) [53].

### 4.1. Glucose and Lipid Metabolism Involve ANGPTL

In the context of glucose metabolism, ANGPTL4 modulates the expression of glucose transporters in colorectal cancer by activating the phosphatidylinositol 3-kinase (PI3K)-protein kinase B (Akt) signaling pathway, thereby promoting glucose metabolism [54]. Conversely, ANGPTL8 reduces hepatic gluconeogenesis in diabetic mice through the activation of the PI3K/Akt pathway, ultimately contributing to the reduction of blood glucose levels [55]. Regarding lipid metabolism, ANGPTL2, ANGPTL3, ANGPTL4, and ANGPTL8 are key regulators. ANGPTL2 interacts with CD146 to activate the cAMP response element-binding protein, leading to the upregulation of CD146 and promoting adipogenesis [56]. ANGPTL3, ANGPTL4, and ANGPTL8 can independently regulate lipid metabolism or act synergistically. Specifically, ANGPTL3 inhibits endothelial cell phospholipase activity [57], while ANGPTL4 induces irreversible unfolding of the α/β-hydrolase domain of lipoprotein lipase (LPL), resulting in LPL inactivation [58]. Both ANGPTL3 and ANGPTL4 possess CCD domains that inhibit LPL function [59]. ANGPTL8 interacts with ANGPTL4 secreted by white adipose tissue, diminishing its function, and binds with ANGPTL3 in serum to further enhance its LPL inhibitory effect [60]. ANGPTL3, ANGPTL4, and ANGPTL8 have been demonstrated to inhibit LPL function via specific epitope 1 [61,62]. In conclusion, ANGPTL4 and ANGPTL8 are implicated in glucose metabolism, while ANGPTL2, ANGPTL3, ANGPTL4, and ANGPTL8 are integral to lipid metabolism, underscoring the significance of ANGPTLs in both glucose and lipid metabolic processes.

### 4.2. Vascular Permeability Promoted by ANGPTL

Numerous studies have indicated that ANGPTL4 promotes vascular leakage. In endothelial cells (EC), ANGPTL4 interacts with intercellular vascular endothelial cadherin and tight junction protein 5 cluster through integrin signaling, thereby compromising the integrity of vascular junctions [63]. Additionally, ANGPTL4 can bind to neuropilin-1 (NRP1) and NRP2 on EC, leading to the rapid activation of the Ras homolog gene family member (Rho) A/Rho-associated coiled-coil forming protein kinase signaling pathway, which disrupts connectivity between EC [64]. Furthermore, research has demonstrated that ANGPTL4 can enhance EC permeability by upregulating VEGF expression levels, an effect that is not entirely VEGF-dependent [65]. ANGPTL4 also ameliorates hypoxia-induced vascular permeability by binding to EC integrin αvβ3, thereby competing for tyrosine kinase Src signaling downstream of VEGFR2 [66]. ANGPTL4 has been shown to mitigate histamine-induced vascular leakage, suggesting a potential protective role in maintaining the integrity of the blood-retinal barrier [67]. In Müller cells, hypoxic conditions activate the hypoxia-inducible factor-1α (HIF-1α) pathway, leading to the upregulation of ANGPTL4, which subsequently enhances vascular permeability [68]. Conversely, in retinal pigment epithelial cells, ANGPTL4 increases blood–retinal barrier permeability by activating signal transducer and activator of transcription 3 (STAT3), resulting in the downregulation of Claudin proteins and zonula occludens-1. Thus, ANGPTL4 exhibits a dual effect on vascular permeability, with its specific mechanisms likely influenced by the cellular environment and its receptor interactions.

### 4.3. Pathological Angiogenesis Involves ANGPTL

Furthermore, ANGPTL2, ANGPTL3, ANGPTL4, and ANGPTL6 are implicated in angiogenesis. Notably, ANGPTL4 promotes ocular angiogenesis by stimulating retinal endothelial cells to express VEGF or profilin-1. Additionally, ANGPTL4 can enhance nitric oxide production through the integrin/Janus kinase/STAT3 pathway, leading to the upregulation of inducible nitric oxide synthase expression in wound epithelium, thereby facilitating angiogenesis [69]. ANGPTL4 has been shown to significantly induce the rupture of the limiting membrane in retinal vessels, thereby promoting the extension of retinal vessels into the vitreous [70]. ANGPTL2, ANGPTL3, and ANGPTL6 are also involved in angiogenesis through distinct mechanisms. Specifically, ANGPTL2 regulates membrane matrix metalloproteinases to control endothelial cell migration and three-dimensional tube formation by activating C-Jun N-terminal kinase (JNK) phosphorylation [71]. ANGPTL3 facilitates neovascularization by binding to the integrin ανβ3 receptor on endothelial progenitor cells, promoting Akt phosphorylation, and upregulating the expression of miR-126 [72]. In patients with diabetic macular edema, the expression level of ANGPTL6 is elevated and positively correlates with disease severity [46]. ANGPTL6 induces angiogenesis by activating the extracellular signal-regulated kinase (ERK) 1/2-endothelial nitric oxide synthase-nitric oxide pathway [73]. In addition to its involvement in the ERK1/2 pathway, ANGPTL6 within exosomes can also facilitate angiogenesis via the JNK and mitogen-activated protein kinase p38 pathways, indicating a potential role for ANGPTL6 in ocular angiogenesis [74].

### 4.4. Intraocular Inflammation Involves ANGPTL

ANGPTL4 has been demonstrated to increase IL-1β and IL-6 levels by triggering the PROFILIN-1 signaling pathway, thus boosting the inflammatory response [66]. Conversely, some studies have demonstrated that ANGPTL4 can inhibit microglial activation and suppress the fatty acid synthase ligand/fatty acid synthase pathway, leading to reduced cell apoptosis and inflammation [75]. This dual functionality suggests that ANGPTL4 may have a complex role in ocular inflammatory lesions. In a mouse model of keratitis, ANGPTL2 expression is upregulated, and it induces macrophage infiltration and IL-1β overexpression, exacerbating corneal inflammatory changes [76]. Furthermore, in a mouse model of uveitis, ANGPTL2 promotes endotoxin-induced retinal inflammation by activating the nuclear factor (NF)-κB signaling pathway [77].

ANGPTL2 activates retinal Müller cells through PirB receptors in a mouse model of LPS-induced endophthalmitis. In a choroidal neovascularization model, ANGPTL2 binds to integrin α4β2, thereby activating the NF-κB and ERK pathways in macrophages. This activation promotes the expression of various inflammatory factors and the recruitment of macrophages, which in turn further enhances ANGPTL2 expression [78]. These findings indicate a strong association between ANGPTL2 and ocular inflammatory changes. Similarly, ANGPTL3 has been shown to directly induce ocular inflammation. Research has demonstrated that intravitreal injection of ANGPTL3 in mice leads to its binding with retinal endothelial cell integrin ανβ3, triggering pro-inflammatory and pro-apoptotic effects. These effects can be inhibited by the peroxisome proliferator-activated receptor α (PPAR-α) agonist fenofibrate [79]. This evidence suggests that ANGPTL3 may facilitate ocular inflammation by suppressing the PPAR-α pathway.

## 5. Therapeutic Implications

ANGPTL-4 potently modulated inflammation, permeability, and angiogenesis through the activation of the profilin-1 signaling pathway. Lu’s findings demonstrated that hyperglycemia (HG) induced ANGPTL-4 upregulation in a HIF-1α-dependent manner, both in vivo and in vitro. These results indicate that targeting ANGPTL-4, either alone or in combination with profilin-1, could serve as an effective therapeutic strategy and potential diagnostic biomarker for proliferative diabetic retinopathy and other vitreoretinal inflammatory diseases [65].

## 6. Discussion

Evidence from research points to ANGPTL3, ANGPTL4, and ANGPTL8 as possible biomarkers for DR. In patients with PDR, the expression of ANGPTL4 in both the vitreous body and serum is notably higher than in patients with IMH. Furthermore, ANGPTL4 and VEGF levels are significantly correlated in the vitreous body and serum of PDR patients, indicating that ANGPTL4 may serve as a new therapeutic target for PDR [6,11]. Research also reveals that serum ANGPTL3 expression is positively related to DR stages in those with T2DM, hinting at its role as a biomarker for DR progression tracking [6]. ANGPTL8 expression is markedly increased in the vitreous body of PDR patients, and its serum levels are significantly associated with DR [8,9]. Consequently, ANGPTL8 may also serve as a diagnostic marker for DR. Currently, three drugs targeting ANGPTL3 are under development: the monoclonal antibody evinacumab, the antisense oligonucleotide vupanorsen, and the RNA interference drug ARO-ANG3. One of these agents, everolimus, is available as an adjuvant therapy for homozygous familial hypercholesterolemia. Research has demonstrated that pharmacological inhibition of ANGPTL3 by evervolumab and Vupanorsen mimics the phenotype associated with ANGPTL3 loss-of-function mutations [80,81]. By targeting circulating ANGPTL3, everolimus has been shown to effectively reduce levels of low-density lipoprotein cholesterol (LDL-C), high-density lipoprotein cholesterol (HDL-C), and TG in both healthy individuals and patients with familial hypercholesterolemia. Notably, the efficacy of evesizumab in populations deficient in LDL receptors (LDLR) indicates that its LDL-C-lowering effect operates independently of functional LDLR [80,82]. However, the requirement for frequent intravenous administration of antibodies may pose challenges to patient convenience and adherence to therapy. Furthermore, the development of Vupanorsen has been halted due to clinical findings of elevated alanine aminotransferase levels and hepatic steatosis [81]. ARO-ANG3, an RNA interference therapy targeting ANGPTL3, is currently undergoing phase II clinical trials. According to the latest clinical trial findings, treatment involving a small interfering RNA aimed at ANGPTL3 mRNA may improve outcomes. The compound was well tolerated by patients and demonstrated efficacy in reducing concentrations of atherosclerotic lipoproteins, TG, and lipoproteins. ARO-ANG3 has also been shown to decrease LDL-C expression levels. This compound addresses a significant gap in the secondary prevention of extensive atherosclerosis and is particularly valuable for managing high-risk populations, such as those with mixed dyslipidemia and familial hypercholesterolemia. However, the development of targeted therapies aimed at ANGPTL in DR has not been reported. Future research should investigate the feasibility of using ANGPTL3, ANGPTL4, and ANGPTL8 as biomarkers in DR and assess whether ANGPTL3-targeted therapies can improve disease progression in DR while managing dyslipidemia. Additionally, the potential of ANGPTL4 as a novel targeted therapy for DR warrants exploration.

## 7. Summary and Prospect

ANGPTL is ubiquitously expressed in the body and has diverse and critical functions. It plays multiple roles, including maintaining glucose and lipid metabolism homeostasis, finely regulating stem cell activity, participating actively in angiogenesis, effectively modulating inflammatory responses, and controlling vascular permeability. ANGPTLs are crucial in maintaining glucose and lipid homeostasis through intricate molecular mechanisms, thereby contributing to the prevention of metabolic disorders. Concurrently, ANGPTLs modulate stem cell activity, influencing tissue repair and regeneration, which is vital for the body’s response to injury and aging.

In terms of angiogenesis, ANGPTLs regulate the proliferation, migration, and differentiation of vascular endothelial cells, significantly impacting neovascularization. This regulation is essential in various physiological and pathological contexts, including embryonic development, wound healing, and tumor progression. Moreover, ANGPTLs participate in regulating inflammatory responses, influencing the activation and movement of immune cells, thereby being crucial in resisting infections, clearing pathogens, and repairing tissues. ANGPTLs play a critical role in the regulation of glucose and lipid homeostasis through complex molecular pathways, thereby aiding in the prevention of metabolic disorders. Additionally, ANGPTLs influence stem cell activity, which is crucial for tissue repair and regeneration, and is essential for the body’s response to injury and the aging process. Moreover, ANGPTLs are implicated in the modulation of inflammatory responses, influencing immune cell activation and migration, and thus play a significant role in infection resistance, pathogen clearance, and tissue repair.

Research conducted afterward has indicated a strong positive correlation between the expression levels of various ANGPTLs in the serum or vitreous body and the onset and progression of DR. This finding indicates that ANGPTLs could serve as biomarkers for predicting the occurrence and advancement of DR. Furthermore, due to their regulatory roles in both normal and pathological processes and their strong association with DR, ANGPTLs are considered potential molecular targets. An in-depth analysis of ANGPTLs could not only clarify the pathogenesis of DR but also introduce new therapeutic approaches. For instance, the development of anti-ANGPTL antibody drugs could disrupt ANGPTL function, thereby inhibiting the progression of DR and offering new prospects for its clinical management.

## Figures and Tables

**Figure 1 metabolites-15-00352-f001:**
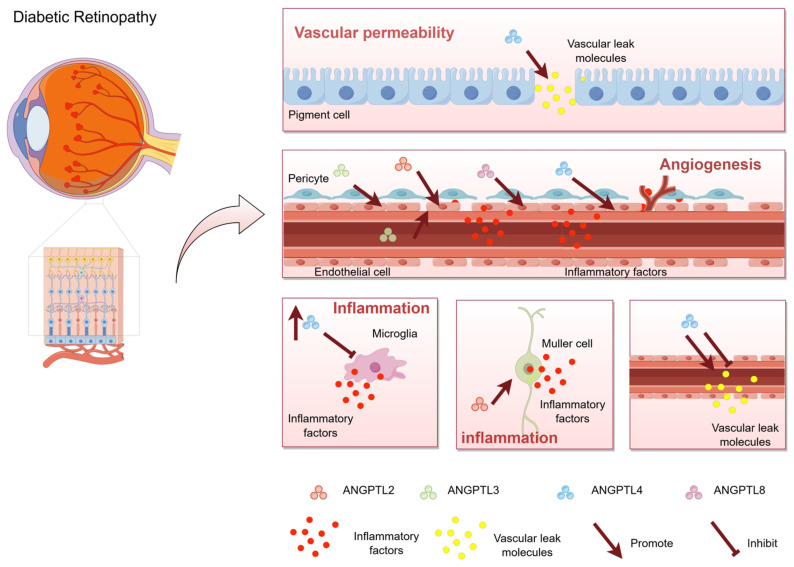
Schematic representation of the role of ANGPTL in diabetic retinopathy. Angiopoietin-like proteins (ANGPTLs) are a group of secreted glycoproteins widely expressed in living organisms and play crucial roles in numerous pathophysiological processes, including glucose and lipid metabolism, stem cell proliferation, local inflammation, vascular permeability, and angiogenesis. The effects of ANGPTL4 on vascular permeability are bidirectional. It can not only promote vascular leakage, but also protect vascular permeability. ANGPTL2, ANGPTL3, ANGPTL4, and ANGPTL8 promote angiogenesis. Among them, ANGPTL2, ANGPTL4, and ANGPTL8 are up-regulated in the eye, which promote endothelial cell migration and induce angiogenesis. ANGPTL3 promotes the differentiation of endothelial progenitor cells in plasma to induce neovascularization. ANGPTL2, ANGPTL3, and ANGPTL4 promote retinal inflammation. Red particles represent inflammatory factors, and yellow particles represent Vascular leak molecules. Arrows represent promotion and flat arrows represent inhibition. This figure was drawn by Figdraw.

**Table 1 metabolites-15-00352-t001:** ANGPTL family protein member.

ANGPTL	Main Tissue Expression	Receptor	Chromosome (Human)	Functional Description	Clinical Significance	Related Diseases	References
ANGPTL1	Liver, muscle, thyroid gland, bladder, gallbladder, gastrointestinal tract (no esophagus), adipose tissue, skin	orphan nuclear receptor, site A apolipoprotein (AI)	1	Promotes angiogenesis, tissue repair, permeability, anti-apoptotic	Potential role in tissue regeneration	Cance, Cardiovascular diseases	[7,10,11,12,13]
ANGPTL2	Heart, vessels, adipose tissue, kidney, lung, skeletal muscle	integrins α5β1 and Toll-like receptor 4 (TLR4), leukocyte immunoglobulin-like receptor B2 (LILRB 2)	9	Inflammation, angiogenesis, development of cancer, regulate lipid metabolism	Biomarker for metabolic disorders	Obesity, Diabetes, Atherosclerosis	[14,15,16,17,18,19]
ANGPTL3	Liver, kidney	alpha-5/beta-3, LILRB2 (weak)	1	Angiogenesis, inhibits Lipoprotein lipase (LPL) activity, regulate lipid metabolism	Therapeutic target for lipid disorders	Hyperlipidemia, Atherosclerosis, Loss-of-function mutations	[20,21]
ANGPTL4	Liver, adipose tissue, brain, intestine, thyroid, kidney, heart, muscle, ovary, testis, kidney urinary bladder, esophagus	fibronectin, vitronectin, integrin β1 and β5	19	Inhibits LPL activity, regulates lipid metabolism, energy balance, angiogenesis, glucose metabolism, redox regulation, inflammation, development of cancer	Biomarker and therapeutic target	Obesity, Diabetes, Cancer	[22,23,24,25,26,27,28,29,30,31]
ANGPTL5	Heart, adipose tissue, ovary, testis, skin	LILRB2	11	Lipid, triglyceride metabolism	—	Cancer	[15,32]
ANGPTL6	Liver, gallbladder, placenta	orphan of receptor	19	Angiogenesis, lipid metabolism, glucose metabolism	Potential role in vascular health	Cardiovascular diseases, Cancer	[16,33,34]
ANGPTL7	Eye (trabecular meshwork)	LILRB2 (weak)	1	Regulates intraocular pressure	Potential therapeutic target for eye diseases	Glaucoma	[15,35,36,37]
ANGPTL8	Liver, adipose tissue	orphan of receptor	19	Regulates insulin sensitivity, lipid metabolism	Therapeutic target for metabolic disorders	Obesity, Type 2 Diabetes	[38,39,40,41,42]

## Data Availability

No new data were created or analyzed in this study.

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
