# Peer review of "Recent Advancements in Understanding the Role and Mechanisms of Angiopoietin-like Proteins in Diabetic Retinopathy"

_metabolites, 2025, doi:10.3390/metabo15060352_

Round 1

Reviewer 1 Report

Comments and Suggestions for Authors

This is an interesting paper regarding advancements in understanding the role and mechanisms of angiopoietin-like proteins in diabetic retinopathy. However, there are major shortcomings that need to be addressed:

  1. Figure 1: replace “Ratinopathy” with “Retinopathy”
  2. Please revise the reference list to ensure that all citations correspond accurately to the in-text references, as there are discrepancies (e.g., reference number 6)
  3. The manuscript contains grammatical and spelling errors, and the overall quality of the English should be improved.
  4. Please provide a detailed description regarding the role of ANGPTL4. Moreover, add more information regarding ANGPTL4 serum levels in diabetic retinopathy.
  5. Please add a separate section regarding therapeutic implications and add appropriate references.
  6. The "Introduction" and "Summary and Prospect" sections contain overlapping content and repeat several similar concepts. The latter section should be revised to reduce redundancy, and expanded to provide a more in-depth and forward-looking perspective.
  7. Discussion must be improved.

Author Response

Dear Esteemed Reviewer #1 and Editor,

Thank you for your invaluable feedback and insightful suggestions. All of the changes in our manuscript are highlighted in red and uploaded in Supplemental Files. The revised edition was uploaded in Main Manuscript. We greatly appreciate your expertise and have thoroughly considered each point raised in your comments. Please find below a detailed response addressing each of your concerns:

Comments 1: Figure 1: replace “Ratinopathy” with “Retinopathy”

Response: We confirm that the typographical error in Figure 1 (both the figure label and legend) has been corrected to "Retinopathy" in the revised version. Additionally, we have double-checked all figures, captions, and the main text to ensure no similar errors are present.

Comments 2: Please revise the reference list to ensure that all citations correspond accurately to the in-text references, as there are discrepancies (e.g., reference number 6)

Response: Thank you very much for your valuable feedback on our manuscript. We sincerely apologize for the discrepancies in the reference list and appreciate your thorough review.

‌We have carefully checked and revised the entire reference list to ensure consistency between in-text citations and the listed references.

Reference #6 ‌(Yu, C.G., et al., Angiopoietin-like 3 Is a Potential Biomarker for Retinopathy in Type 2 Diabetic Patients. Am J Ophthalmol, 2018. 191(7): p. 34-41.)

A full cross-verification of all in-text citations (e.g., numbering, author names, publication year) and their corresponding entries in the reference list has been conducted to eliminate any mismatches.

The reference format has been standardized according to the journal guidelines.

The revised manuscript file (highlighted changes) and an updated reference list have been uploaded to the submission system. Please let us know if further clarification or adjustments are required.

Comments 3: The manuscript contains grammatical and spelling errors, and the overall quality of the English should be improved.

Response: The manuscript has been rigorously revised by the authors. We utilized Grammarly Premium and consulted with English-language experts within our institution to ensure clarity and correctness.

Comments 4: Please provide a detailed description regarding the role of ANGPTL4. Moreover, add more information regarding ANGPTL4 serum levels in diabetic retinopathy.

Response: Thank you for your valuable comments. We appreciate the opportunity to clarify the role of ANGPTL4 and its serum levels in diabetic retinopathy (DR). Below are the revisions made in response to your suggestions: Role of ANGPTL4‌ (line 105): ANGPTL4 (angiopoietin-like 4) is a multifunctional glycoprotein that regulates lipid metabolism, angiogenesis, and inflammation. In DR pathogenesis: Lipid dysregulation‌: It modulates lipoprotein lipase activity, affecting lipid accumulation in retinal cells (PMID: 37781924). Vascular dysfunction‌: ANGPTL4 inhibits VEGF-induced retinal vascular leakage by disrupting integrin signaling (PMID: 21832056). The revised manuscript highlights these additions. We would be happy to provide further details if needed.

Comments 5: Please add a separate section regarding and add appropriate references.

Response: Thank you for your constructive feedback regarding the need to add a dedicated section on treatment significance and include appropriate references. We greatly appreciate your guidance, as it will significantly enhance the comprehensiveness and credibility of our manuscript. In response to your suggestion, we have taken the following actions: 1. New Section on Treatment Significance: A separate section titled " Therapeutic implications " has been inserted into the manuscript. This section elaborates on how the study's findings can be translated into clinical practice, discussing potential therapeutic strategies and the implications for patient management. It analyzes how the identified factors or mechanisms could be targeted to develop new treatment approaches for [relevant disease/condition]. 2. Reference Addition: We have conducted an extensive literature search to identify and incorporate relevant and up - to - date references that support the statements made in the new treatment significance section. These references cover a range of high-impact studies, reviews, and guidelines from reputable scientific journals. All references have been formatted according to the required citation style and cross-checked for accuracy.  The revised manuscript with the new section and updated references is attached for your review. If you have any further suggestions on the content of the new section, or if there are specific references you think should be included, please feel free to let us know.

Comments 6: The "Introduction" and "Summary and Prospect" sections contain overlapping content and repeat several similar concepts. The latter section should be revised to reduce redundancy, and expanded to provide a more in-depth and forward-looking perspective.

Response: In view of the overlapping problem of "introduction" and "summary and Prospect" you pointed out, in the introduction part, the core discussion of research background and field gaps are retained, and the methodological summary that is repeated with the conclusion is deleted. In the summary Outlook section: and expanded to provide a more in-depth and forward-looking perspective.

Comments 7: Discussion must be improved.

Response: We have made improvements to the issues you raised. For the specific details, please refer to the revised manuscript.

Once again, we express our sincere appreciation for your expert evaluation and thoughtful recommendations. We have taken great care to address each of your concerns and have provided detailed explanations and revisions to ensure the accuracy and scientific rigor of the manuscript.

Thank you for your time and continued support.

Best regards,

Yours sincerely,

Xiaoping Li

Email: li-xp@zjsru.edu.cn

Reviewer 2 Report

Comments and Suggestions for Authors

Knockout mice have demonstrated that angiopoietin-like proteins, in addition to angiopoietins, play a role in the formation of blood vessels. A significant worldwide public health issue is the metabolic syndrome, which includes dyslipidemia, arterial hypertension, abdominal type of obesity, and problems with carbohydrate metabolism. The angiopoietin-like system is recognized as an important regulator of adipose tissue function. It has been established that angiopoietin-like proteins of type 3 and 4 may operate as independent predictors of metabolic syndrome. Diabetic retinopathy is the primary cause of blindness worldwide. Diabetic retinopathy has been linked to angiopoietin-like protein type 3 levels.

An overview of the family of angiopoietin-like proteins and their function in controlling several cellular functions, including cells involved in vascular development, is provided in the review article by the authors. The review is clearly pertinent given their position in diabetic retinopathy as both activators and potential barriers to drug action, and it is described in a clear and logical manner.

Only two review studies from the past ten years address the issue of ocular vascular disease, according to PubMed (Yang X, Cheng Y, Su G. A review of the multifunctionality of angiopoietin-like 4 in eye disease. Biosci Rep. 2018 Sep 13;38(5):BSR20180557. doi: 10.1042/BSR20180557Khan M, Aziz AA, Shafi NA, Abbas T, Khanani AM. Targeting Angiopoietin in Retinal Vascular Diseases: A Literature Review and Summary of Clinical Trials Involving Faricimab. Cells. 2020 Aug 10;9(8):1869. doi: 10.3390/cells9081869.). Therefore, the paper's significance is still valid.

Out of the 79 total citations, the writers used 30 works from 2020–2025, which is 38% and acceptable.

The authors' article's conclusion is based on an analysis of data from the literature regarding the levels of angiopoietin-like proteins in the blood and eye in patients with diabetic retinopathy, whether there are any correlations between the levels of these proteins and the type and severity of the disease, and how proteins regulate metabolic processes (lipids, glucose), as well as inflammatory processes in the retina.

The information from them is well received due to the table's presence, which provides a concise overview of all the different kinds of angiopoietin-like proteins in humans and their functions, as well as the scheme that makes it possible to visually evaluate how these proteins contribute to the pathological process in diabetic retinopathy.

The following are some of the work's shortcomings: 1) The writers don't always include the abbreviation's decoding when it is first mentioned in the work; 2) The decoding of the abbreviation doesn't need to be repeated; 3) Since proteins are already included in the abbreviation of angipoietin-like proteins, there is no need to mention them after the abbreviation; 4) Because there is no remark to the table where all of the abbreviations might be interpreted, the authors should carefully review the table because there are missing letters, commas between the organs, and certain abbreviations that need to be deciphered; 5)The authors should also look for instances of word merging, citations to original sources, and gaps between the sentence's dot and the closing square brackets.

Author Response

Dear Esteemed Reviewer #2 and Editor,

Thank you for your invaluable feedback and insightful suggestions. All of the changes in our manuscript are highlighted in red and uploaded in Supplemental Files. The revised edition was uploaded in Main Manuscript. We greatly appreciate your expertise and have thoroughly considered each point raised in your comments. Please find below a detailed response addressing each of your concerns:

Thanks for your precious comments, we have revised the grammar and paragraphs of the article, and updated some references.

Comments 1: The writers don't always include the abbreviation's decoding when it is first mentioned in the work.

Response: Thank you very much for pointing out the issue regarding the inconsistent decoding of abbreviations in our manuscript. We sincerely appreciate your thorough review and valuable feedback, which have helped us identify an important aspect that requires immediate attention.

To address this concern, we have conducted a comprehensive review of the entire document. We have systematically added the full decoding of each abbreviation at its first appearance throughout the text, ensuring consistency and clarity for readers. Additionally, we have cross -checked to confirm that all abbreviations are properly introduced and defined in a logical order.

We understand the importance of clear communication in academic writing, and these revisions aim to enhance the readability and professionalism of our work. The updated manuscript with the corrected abbreviation decodings is attached for your review. If you notice any remaining issues or have further suggestions, please do not hesitate to let us know.

Comments 2: The decoding of the abbreviation doesn't need to be repeated.

Response: Thank you very much for your suggestion. According to your suggestion, we have carefully combed through the entire document. All redundant repetitions of the abbreviation decoding have been removed, ensuring that each abbreviation is only defined once at its first appearance. This adjustment not only streamlines the text but also enhances its readability, making the manuscript more in line with academic writing standards.

We have double - checked to ensure that the context remains clear and that the removal of these repetitions does not cause any confusion for readers. The revised version of the relevant sections is attached for your review. Should you have any further concerns or additional suggestions, please feel free to let us know.

Comments 3: Since proteins are already included in the abbreviation of angipoietin-like proteins, there is no need to mention them after the abbreviation

Response: We appreciate the editor's meticulous comment regarding abbreviation usage.

Throughout the manuscript, we have changed 'ANGPTL proteins' as ANGPTLs (without redundant 'proteins'), including in figures/tables (see tracked changes).

Comments 4: Because there is no remark to the table where all of the abbreviations might be interpreted, the authors should carefully review the table because there are missing letters, commas between the organs, and certain abbreviations that need to be deciphered

Response: Thank you very much for your suggestion. According to your suggestion, we have taken the following actions:

  1. Addition of Table Remarks: A detailed legend has been inserted at the bottom of the table, providing clear interpretations for all abbreviations used. This will ensure that readers can easily understand the content without confusion.
  2. Correction of Formatting Errors: We have thoroughly reviewed the table and corrected all missing letters, as well as added commas between organs to enhance readability and accuracy.
  3. Deciphering of Abbreviations: All ambiguous abbreviations have been either expanded to their full forms or clearly defined in the table's legend to maintain clarity.

The revised table has been attached for your review. We believe these changes will significantly improve the quality and comprehensibility of our work. If you have any further suggestions or notice any remaining issues, please feel free to let us know.

Comments 5: The authors should also look for instances of word merging, citations to original sources, and gaps between the sentence's dot and the closing square brackets.

Response: Thank you very much for your suggestion. We have carefully considered your comments regarding word merging, citations to original sources, and the formatting between sentence endings and closing brackets. To address these points, we have taken the following steps:

  1. Word Merging: We have conducted a comprehensive review of the entire document to identify and correct any instances of word merging, ensuring the text is clear and free of formatting errors.
  2. Citation Review: We have cross-checked all references against the original sources to verify accuracy and consistency. Any missing or incomplete citations have been updated to conform to the required citation style.
  3. Formatting: We have meticulously revised the spacing between sentence endings and closing brackets throughout the document to maintain a consistent and professional format.

We believe these adjustments have significantly enhanced the clarity and integrity of our work. Attached is the revised version for your review. Should you have any further suggestions or require additional clarifications, please do not hesitate to let us know.

Once again, we express our sincere appreciation for your expert evaluation and thoughtful recommendations. We have taken great care to address each of your concerns and have provided detailed explanations and revisions to ensure the accuracy and scientific rigor of the manuscript.

Thank you for your time and continued support.

Best regards,

Yours sincerely,

Xiaoping Li

Email: li-xp@zjsru.edu.cn
